# Integrated Analysis of Transcriptome and Metabolome Reveals New Insights into the Molecular Mechanism Underlying the Color Differences in Wolfberry (*Lycium barbarum*)

Linyuan Duan, Bo Zhang, Guoli Dai, Xinru He, Xuan Zhou, Ting Huang, Xiaojie Liang, Jianhua Zhao * and Ken Qin *

Institute of Wolfberry Science, Ningxia Academy of Agriculture and Forestry Sciences, Yinchuan 750002, China; dly698013@163.com (L.D.); zhangbo0309@163.com (B.Z.); dgl2006swfc@163.com (G.D.); hhexinru@163.com (X.H.); m18695145633@163.com (X.Z.); ht20180312@163.com (T.H.); lxj910303@126.com (X.L.)
* Correspondence: zhaojianhua0943@163.com (J.Z.); qinken7@163.com (K.Q.)

**Abstract:** Wolfberry (*Lycium barbarum* L.) is a small Solanaceae shrub with medicinal and edible homology, and widely used as ethnobotanical medicine and nutraceutical food. The wolfberry fruits mainly have red, purple, and yellow phenotypes. Wolfberries are rich in flavonoids, which are natural water-soluble pigments that endow a variety of colors in plants There are very few investigations on mechanism of flavonoids biosynthesis and fruit coloring reported about wolfberry. The widely targeted metabolome and transcriptome analysis were performed to obtain metabolite and gene expression profiles of red, yellow, and purple wolfberries and to explain the underlying molecular mechanism of the color differences in wolfberry. As result, metabolomics analysis revealed that the bluish anthocyanins Malvidin and petunidin trended to accumulate in purple wolf-berry, while red and yellow wolfberries trended to accumulate more yellowish flavonoids. And transcriptome analysis showed that flavonoid synthesis-related genes, such as *CHS*, *F3H*, *ANS* and *DFR*, and several *MYB* and *bHLH* genes were differentially expressed among wolfberries in different colors: most of them were more highly expressed in purple wolfberries than in red and yellow ones. In conclusion, the different flavonoids' accumulation patterns may result in the different fruit colors of wolfberry, and the MYB or bHLH transcription factors could regulate the expression of flavonoids biosynthesis related genes to change the composition of flavonoids or anthocyanins in wolfberry fruits and result in varied fruit colors. These findings provide new insights into the underlying molecular mechanism of the fruit color differences in wolfberry and provide new ideas for molecular breeding of wolfberry.

**Keywords:** metabolomics; transcriptomics; flavonoid biosynthesis; anthocyanin biosynthesis

## 1. Introduction

Wolfberry (*Lycium barbarum* L.) is a small Solanaceae shrub with medicinal and edible homology, and widely used as ethnobotanical medicine and nutraceutical food. It has been implicated in replenishing vital essence, improving eyesight, and nourishing the liver and kidneys [1,2]. Pharmacological studies of wolfberry compounds have shown many beneficial uses. For instance, *L. barbarum* polysaccharides (LBPs), unique, water-soluble glycoconjugates in wolfberry, possess antiaging, antidiabetic, antifibrotic, neuroprotective, and immunomodulating properties [3]. Furthermore, polyphenols such as flavonols, anthocyanins, and catechins have been found to be strong natural antioxidants [1].

Color has been used as a quality evaluation index in traditional Chinese medicine and is related to its intrinsic material base. Common plant pigments include chlorophyll, carotenoid, beet pigment, and flavonoid pigment, the physical and chemical properties of which, as well as mutual mixing and modulation, create colorful color characteristics of plants. Anthocyanins, a subgroup of flavonoids, are natural, water-soluble pigments that endow a variety of colors in plants [4–6]. Six main anthocyanin pigments in plants, such

as cyanidin, delphinidin, pelargonin, peonidin, malvidin, and petunidin, show different colors [7,8]. Thus, the anthocyanin biosynthesis pathway is related to color phenotype [4–6].

Currently, much is known about the core flavonoid biosynthesis pathway. Firstly, the phenylalanine ammonia lyase (PAL), cinnamate-4-hydroxylase (C4H), and 4-coumarate CoA ligase (4CL) convert phenylalanine to coumaroyl-CoA. Secondly, coumaroyl-CoA and malonyl-CoA are synthesized into dihydroflavonol with catalyticases such as chalcone synthase (CHS), chalcone isomerase (CHI), flavanone 3-hydroxylase (F3H), flavonoid 30-hydroxylase (F30H), flavonol synthase (FLS), and flavone synthase (FNS). Thirdly, dihydroflavonol forms leucoanthocyanidins via the action of dihydroflavonol 4-reductase (DFR), and then anthocyanidins are synthesized by anthocyanidin synthase (ANS). Finally, the anthocyanidins are modified by flavonoid glucosyltransferase (UFGT) and anthocyanin O-methyltransferase (AOMT) [9–11].

The integrated analysis of transcriptomics and metabolomics can comprehensively reflect the nature of sample phenotypic differences by comparing the differences at the gene and metabolite level of samples with different phenotypes, combining the relationship between the expression changes of these two substances [12,13], and have been widely used to study the synthesis and regulation mechanisms of metabolites in many plants [5,6], such as tomato fruits [14], watermelon fruits [15], peach fruits [16], and sugarcane [17]. There have been limited investigations on the regulatory mechanisms of anthocyanin synthesis of red, yellow, and purple fruit colors in wolfberry [18].

Hence, in order to determine the composition patterns of flavonoid compounds and the flavonoid-synthesis-related gene expression profiles in wolfberry, widely targeted metabolome and transcriptome analysis of red, yellow, and purple wolfberries was performed and integrated. It is vital for exploring the relationship between gene expression, metabolite accumulation and color formation, and explaining the underlying molecular mechanism of the color differences in wolfberry. And the results would provide new ideas and a new theoretical basis for the molecular breeding of wolfberry.

## 2. Materials and Methods

### 2.1. Plant Materials

The fruits of the red wolfberry cultivar, Ningqi7, yellow wolfberry cultivar, W-12-27, and purple wolfberry cultivar, ZH-13-01, were collected from the germplasm resource nursery of *Lycium barbarum* in Luhuatai Garden Farm, Yinchuan City, Ningxia in 2019. The matured wolfberry fruits in different colors were frozen in liquid nitrogen, and then stored at $-80\ ^\circ$C for subsequent metabolome and transcriptome analysis. All experimental samples were repeated three times.

### 2.2. Metabolites Extraction

About 100 mg of the frozen fruit tissue was crushed with a zirconia bead in a mixer mill (MM 400, Retsch) for 1.5 min at 30 Hz and extracted with 0.6 mL 70% aqueous methanol for overnight at 4 $^\circ$C. Prior to LC-MS analysis, the extracts were centrifugated at $10,000\times g$ for 10 min and then filtrated using microporous membranes (SCAA-104, 0.22 μm pore size; ANPEL, Shanghai, China).

### 2.3. LC-ESI-MS/MS Analysis

The HPLC system (Shim-pack UFLC SHIMADZU CBM30A) with Waters ACQUITY UPLC HSS T3 C18 HPLC column (1.8 μm, 2.1 mm $\times$ 100 mm) and the LC-ESI-MS/MS system (Applied Biosystems 4500 Q TRAP) were utilized for metabolomics analysis of the sample extracts. And the mobile phase solvent A was 0.1% formic acid in water and solvent B was acetonitrile with 0.1% formic acid. The chromatography flow rate was 350 μL/min and a linear gradient was programmed for 14 min: 95%A:5%B (*v*/*v*) at 0 min, 5%A:95%B at 9.0 min, 5%A:95%B at 10.0 min, 95%A:5%B at 11.1 min and kept for 2.9 min. And the column oven temperature was 40 $^\circ$C and the injection volume was 2 μL [19].

Linear ion trap (LIT) and triple quadrupole (QQQ) scans were obtained with a triple quadrupole-linear ion trap mass spectrometer (Q TRAP), API 4500 Q TRAP LC/MS/MS System, equipped with an ESI Turbo Ion-Spray interface, operating in positive and negative ion mode and controlled via Analyst 1.6.3 software (AB Sciex) [19]. The ESI source operation parameters were as follows: ion source, turbo spray; source temperature 550 °C; ion spray voltage (IS) 5500 V (positive ion mode)/−4500 V (negative ion mode); ion source gas I, gas II, and curtain gas were set at 50, 60, and 30.0 psi, respectively [19]. The collision gas was high. Instrument tuning and mass calibration were performed with 10 and 100 µmol/L polypropylene glycol solutions in QQQ and LIT modes, respectively. QQQ scans were obtained as MRM experiments with collision gas (nitrogen) set to 5 psi [19]. DP and CE for individual MRM transitions was done with further DP and CE optimization. A specific set of MRM transitions were monitored for each period according to the metabolites eluted within this period [19].

### 2.4. Metabolite Data Analysis

Before metabolite data analysis, a quality control (QC) analysis was conducted to confirm the reliability of the data. The QC sample, the mixture of all sample extracts, was used to monitor the changes in repeated analyses. Metabolite identification was performed basing on the Metware MWDB database, following their standard metabolic operating procedures [19]. Sciex analyst work station software Version 1.6.3 (Sciex, Framingham, MA, USA) was employed for multiple-reaction-monitoring (MRM) metabolite quantification [20]. The principal component analysis (PCA), partial least squares discriminant analysis (PLS-DA), and 200 permutation tests for PLS-DA were performed to obtain the variable importance in projection (VIP). Metabolites with VIP $\geq 1$ and fold change $\geq 2$ or fold change $\leq 0.5$ were considered as the significantly differentially accumulated metabolites (DAM). Additionally, the Pearson correlation analysis and hierarchical clustering heatmap were performed with R software version 4.2.1 (www.r-project.org, accessed on 21 September 2022).

### 2.5. Transcriptome Sequencing and Data Analysis

The TRIzol reagent (Invitrogen, Waltham, CA, USA) was used to extract the total RNA of samples according to the manufacturer's instructions. The RNase-Free DNase (Promega, Madison, WI, USA) was used to remove genomic DNA in the RNA extracts. The concentration and purification of the RNA was estimated via Qubit 2.0 fluorometer (Life Technologies, Carlsbad, CA, USA) and Agilent Bioanalyzer 2100 (Agilent Technologies, Palo Alto, CA, USA), and then the RNA quality was confirmed via 1% agarose gel electrophoresis. The cDNA libraries were constructed with the eligible RNA extracts. Then the high-throughput sequencing was carried out in Metware Biotechnology Co., Ltd. (Wuhan, China) on the Illumina HiSeqTM 2500 platform (Illumina Inc., San Diego, CA, USA) [15]. Subsequently, after removing the adaptors and the low-quality reads, the clean reads were obtained and then mapped to the reference *Lycium barbarum* genome sequence [21] via HISAT2 v2.1.0. And the gene expression quantification was performed based on the fragments per kilobase of exon model per million mapped fragments (FPKM) using featureCounts v1.6.1. The DESeq2 (version 1.40.2) package was used to perform the differential gene expression analysis with R version 4.2.1 [22]. The genes whose fold change was $\geq 2$ or $\leq 0.5$, and whose Student's T test *p* value was <0.05 and whose Q value was <0.05 were regarded as the differentially expressed genes (DEGs). The Gene Ontology (GO) and Kyoto Encyclopedia of Genes and Genomes (KEGG) pathway functional enrichment analyses of DEGs were performed, respectively. Fisher's exact test was used to evaluate the significance of the GO and KEGG enrichment analyses.

## 3. Results

### 3.1. Transcriptome and Metabolome Characteristics of L. barbarum Fruits Colors

To better understand the molecular basis of yellow, red, and purple fruit colors in *L. barbarum*, fruit samples were collected (Figure 1), and their metabolomes and transcriptomes were analyzed using widely targeted metabolomics and high-throughput RNA-sequencing technologies. Based on these results, principal component analysis (PCA) and Pearson correlation analysis were performed. The PCA results showed that the yellow, red, and purple fruits were clustered in different PCA spaces, both in gene expression level and metabolite abundance level, indicating that the differently colored *L. barbarum* fruits had significant differences in gene expression and metabolite accumulation. Thus, the molecular characteristics may be responsible for the different color phenotypes (Figure 2A). The Pearson correlation analysis showed that the transcriptome correlation between yellow and purple fruits was higher than that between red and yellow or purple fruits. Although the metabolome PCA showed that the different colors were distinguishable, the correlation between their metabolome was high, indicating robust reproducibility of the metabolomics and transcriptomic experiments (Figure 2A).

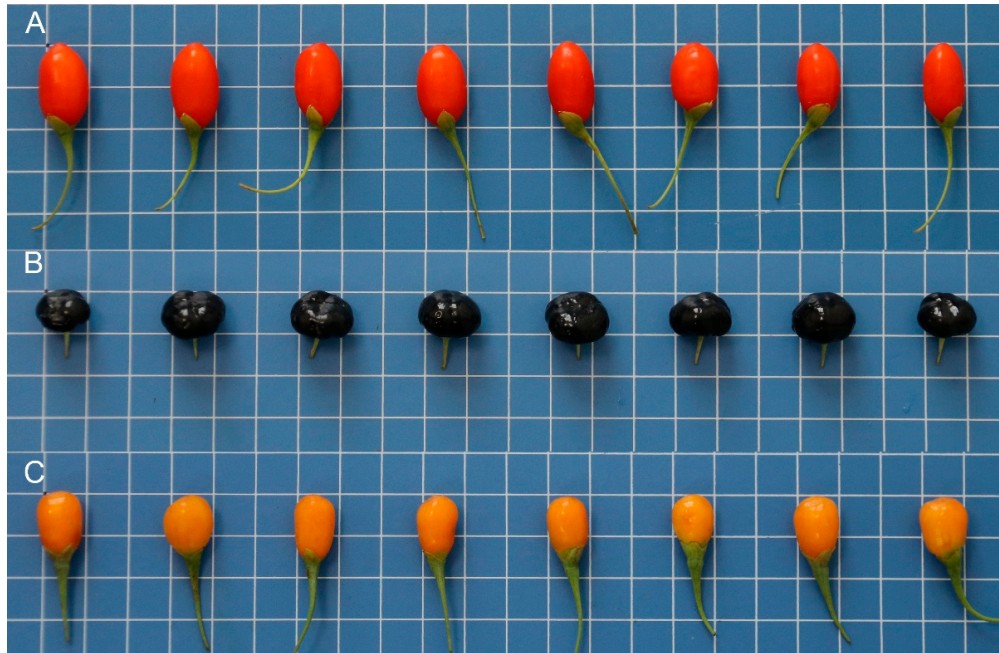

**Figure 1.** Fruits of the yellow, red and purple wolfberry. (**A–C**) are the red, purple, and yellow wolfberry fruits, respectively.

### 3.2. Differentially Expressed Genes among Yellow, Red, and Purple Lycium barbarum Fruits

The transcriptome sequencing of yellow, red, and purple *L. barbarum* fruits generated 64.63 Gb of clean reads after eliminating the adaptor sequences and low-quality reads. The average percentage of the high-quality score (Q30) was 97.45%, the GC contents were 42.5%, and the average mapped ratio of clean reads was 89.66% (Table 1). There were 5788 (1889 up-regulated, 2146 down-regulated), 4462 (2614 up-regulated, 3174 down-regulated), and 4035 (1947 up-regulated, 2515 down-regulated) differentially expressed genes (DEGs) that were screened out in the criterion whose FC was >2 and *p*-value was <0.05 in R_vs._Y, P_vs._Y, and R_vs._P comparison groups, respectively (Table S1; Figure 3A). Among these comparison groups, there were 752 common DEGs (Figure 3B). Furthermore, we found that some genes of the enzymes involved in the core flavonoid biosynthesis pathway were differentially expressed between the three different wolfberries, such as phenylalanine ammonia-lyase (PAL), chalcone isomerase (CHI), chalcone synthase (CHS), flavanone 3-hydroxylase (F3H), flavonoid 3-glucosyl transferase (UFGT), 4-coumarate—CoA ligase

(4CL), anthocyanidin synthase (ANS), and anthocyanin O-methyltransferase (AOMT) (Table 2). Most of PAL, CHS, CHI, F3H, UFGT, ANS, and AOMT genes were more highly expressed in purple wolfberries than in red and yellow wolfberries (Table 2). Meanwhile, a lot of transcription-factor-coding genes were found to be differentially expressed between the three different wolfberries, like MYB and bHLH. As shown in Table 2, we found that most of the MYB and bHLH genes' expression levels were higher in purple wolfberries than that in red and yellow wolfberries (Table 2).

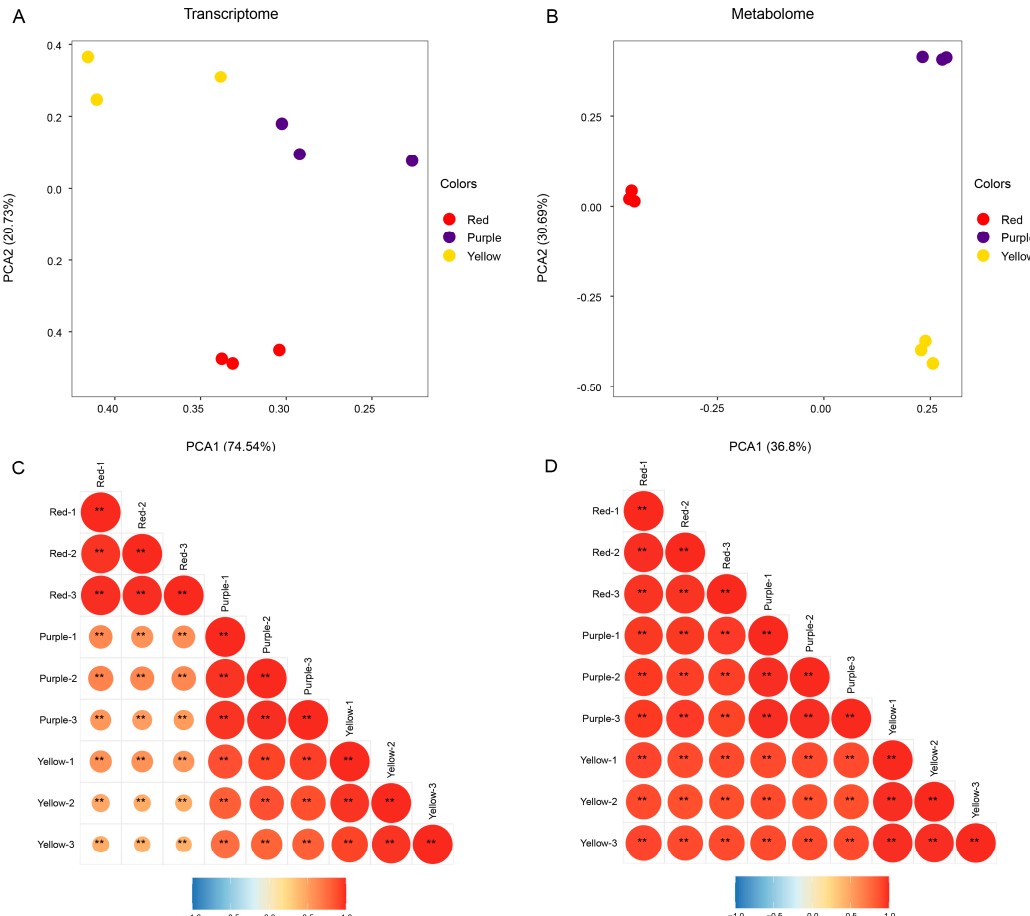

**Figure 2.** Transcriptome and metabolome characteristics of yellow, red and purple wolfberry fruits. (**A**,**B**) Are the PCA results of transcriptome and metabolome, respectively. Red, purple and yellow fruit samples in red, purple and yellow, respectively. (**C**,**D**) Are the Pearson correlation analysis results of transcriptome and metabolome, respectively. "**" refer to Pearson correlation coefficients significance levels <0.01 and <0.001, respectively. The scale color from blue to red indicates the Pearson correlation coefficients from small to large.

**Table 1.** Summary of transcriptomic data.

| Sample | Raw Read | Clean Read | Q20% | Q30% | GC% |
|---|---|---|---|---|---|
| Red_1 | 63,079,672 | 49,752,022 | 99.98 | 97.23 | 42.50 |
| Red_2 | 56,297,936 | 42,361,522 | 99.98 | 97.13 | 42.50 |
| Red_3 | 63,970,362 | 44,320,914 | 99.98 | 97.70 | 42.50 |
| Purple_1 | 53,299,500 | 51,616,046 | 99.97 | 97.47 | 42.50 |
| Purple_2 | 60,853,292 | 59,079,736 | 99.97 | 97.78 | 42.50 |
| Purple_3 | 68,764,014 | 50,181,666 | 99.97 | 97.47 | 42.50 |
| Yellow_1 | 53,716,626 | 41,536,442 | 99.95 | 97.29 | 42.50 |
| Yellow_2 | 56,895,446 | 38,520,522 | 99.95 | 97.40 | 42.50 |
| Yellow_3 | 67,313,950 | 53,530,618 | 99.96 | 97.60 | 42.50 |

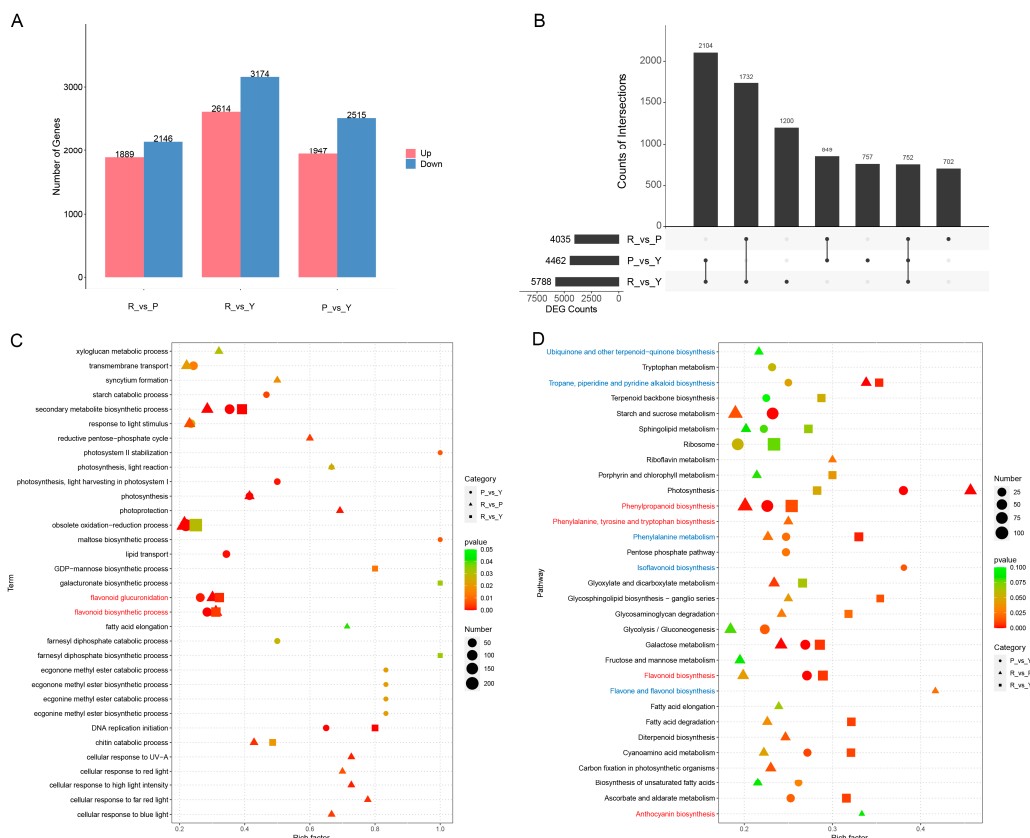

**Figure 3.** Statistical and functional enrichment analysis of differentially expressed genes. (**A**) The barplot of DEGs among yellow, red, and purple wolfberries. (**B**) The upset plot of DEGs. (**C**) The babble plot of GO biological process enrichment analysis; the circle, triangle, and square stands for R_vs._Y, R_vs._P and P_vs._Y comparison groups, respectively. (**D**) The babble plot of KEGG enrichment analysis; the circle, triangle and square stands for R_vs._Y, R_vs._P and P_vs._Y comparison groups, respectively.

**Table 2.** The DEGs of key enzymes of the core flavonoid biosynthesis pathway and related transcription factors.

| | Gene ID | Gene Name | Description | FPKM Expression Mean | | |
| --- | --- | --- | --- | --- | --- | --- |
| | | | | Red | Purple | Yellow |
| Enzymes | CCG041018 | ANS | anthocyanidin synthase | 2.22 | 995.04 | 4.40 |
| | CCG036272 | AOMT | anthocyanin O-methyltransferase | 0.13 | 1863.70 | 6.25 |
| | CCG005771 | CHI | chalcone isomerase | 5.41 | 310.74 | 7.38 |
| | CCG033659 | F3H | flavanone 3-hydroxylase | 72.23 | 3110.26 | 129.96 |
| | CCG048821 | CHI | chalcone isomerase | 2.51 | 100.35 | 4.05 |
| | CCG011047 | CHS | chalcone synthase, partial | 0.10 | 109.61 | 2.18 |
| | CCG025288 | CHS | chalcone synthase, partial | 7.20 | 657.56 | 27.33 |
| | CCG040770 | 4CL | 4-coumarate-CoA ligase 2 | 4.69 | 27.16 | 3.91 |
| | CCG014669 | PAL | phenylalanine ammonia-lyase | 2.37 | 46.59 | 6.15 |
| | CCG027374 | 4CL | 4-coumarate-CoA ligase 1 | 0.06 | 1.13 | 0.15 |
| | CCG021038 | UFGT | flavonoid 3-glucosyl transferase precursor | 8.34 | 235.24 | 23.76 |
| | CCG032840 | 4CL | 4-coumarate-CoA ligase 1 | 10.37 | 21.27 | 38.65 |
| | CCG021591 | 4CL | 4-coumarate-CoA ligase-like 6 isoform X2 | 2.24 | 4.49 | 4.79 |
| | CCG042047 | 4CL | 4-coumarate-CoA ligase 2 | 1.31 | 1.42 | 3.34 |
| | CCG019697 | F3H | flavonol synthase/flavanone 3-hydroxylase-like | 6.88 | 16.41 | 31.92 |
| | CCG014668 | PAL | phenylalanine ammonia-lyase | 2.33 | 5.94 | 3.13 |
| | CCG019697 | FLS | flavonol synthase/flavanone 3-hydroxylase-like | 6.88 | 16.41 | 31.92 |
| | CCG028824 | DFR | dihydroflavonol-4-reductase | 0.42 | 528.47 | 2.07 |
| | CCG031097 | ANR | anthocyanidin reductase-like | 4.06 | 9.64 | 7.16 |

**Table 2.** *Cont.*

| | Gene ID | Gene Name | Description | FPKM Expression Mean | | |
|---|---|---|---|---|---|---|
| | | | | Red | Purple | Yellow |
| Transcription factors | CCG024292 | MYB | MYB1 | 0.37 | 139.59 | 1.16 |
| | CCG005510 | MYB | MYBST1 protein | 59.06 | 264.58 | 161.73 |
| | CCG031911 | MYB | transcription factor MYB1R1 | 33.73 | 122.07 | 79.18 |
| | CCG044025 | MYB | transcription factor MYB78-like | 4.99 | 1.37 | 6.19 |
| | CCG047101 | MYB | transcription factor MYB48-like | 2.89 | 9.27 | 6.70 |
| | CCG015627 | MYB | transcription factor MYB48-like | 0.97 | 3.61 | 6.02 |
| | CCG037979 | MYB | Transcription factor MYB39 | 0.07 | 0.82 | 0.62 |
| | CCG043970 | MYB | transcription factor MYB108-like | 1.91 | 0.76 | 3.70 |
| | CCG028531 | MYB | transcription factor MYB41-like | 0.78 | 1.80 | 0.36 |
| | CCG030977 | MYB | transcription factor MYB101-like | 0.36 | 0.95 | 0.98 |
| | CCG028535 | MYB | transcription factor MYB41-like | 0.52 | 1.22 | 0.37 |
| | CCG036054 | bHLH | transcription factor bHLH66 | 1.56 | 28.54 | 18.99 |
| | CCG028743 | bHLH | transcription factor bHLH13-like | 1.15 | 9.39 | 2.30 |
| | CCG033626 | bHLH | transcription factor bHLH36-like | 0.00 | 1.04 | 0.05 |
| | CCG014150 | bHLH | transcription factor bHLH94-like | 0.48 | 4.94 | 1.51 |
| | CCG011083 | bHLH | transcription factor bHLH148-like | 61.29 | 168.73 | 50.65 |
| | CCG042659 | bHLH | transcription factor bHLH137-like | 4.25 | 14.98 | 7.53 |
| | CCG050226 | bHLH | transcription factor bHLH153-like | 28.62 | 60.52 | 70.14 |
| | CCG041157 | bHLH | transcription factor bHLH62-like | 4.69 | 9.49 | 1.93 |
| | CCG033109 | bHLH | transcription factor bHLH30-like | 0.32 | 1.90 | 5.21 |
| | CCG047551 | bHLH | transcription factor bHLH117 | 0.99 | 3.37 | 6.48 |

*3.3. Functional Enrichment Analysis of the DEGs among Yellow, Red, and Purple Wolfberry Fruits*

In order to clarify the functional category of DEGs and the biological pathways involved, all DEGs were annotated with the Gene Ontology (GO) and the Kyoto Encyclopedia of Genes and Genomes (KEGG) database, and functional enrichment analysis was performed (Figure 3C,D; Tables S2 and S3). The GO enrichment results showed that the DEGs of the R_vs._Y, R_vs._P, and P_vs._Y comparison groups were commonly enriched in the secondary metabolite biosynthetic process, heat response, obsolete oxidation—reduction process, and flavonoid-associated biological processes such as flavonoid glucuronidation. The DEGs of P_vs._Y comparison groups were specifically enriched in toxin catabolic processes, starch catabolic processes, response to oxidative stress, reactive oxygen species metabolic processes, glutathione metabolic processes, photosystem II stabilization (photosynthesis light harvesting unit in photosystem I), and lipid transport. The R_vs._P DEGs were specifically enriched in xyloglucan metabolic processes, syncytium formation, response to Karrikin, reductive pentose—phosphate cycle, photoprotection, cellular response to UV-A, cellular response to red light, cellular response to high light intensity, and cellular response to far red light. The DEGs of R_vs._Y were specifically enriched in cold response and processes, including GDP—mannose biosynthesis, galacturonate biosynthesis, farnesyl diphosphate biosynthesis, and abscisic acid biosynthesis (Figure 3C; Table S2).

The KEGG pathway enrichment results showed that pathways such as tropane, piperidine and pyridine alkaloid biosynthesis, sphingolipid metabolism, photosynthesis, phenylpropanoid biosynthesis, phenylalanine metabolism, galactose metabolism, flavonoid biosynthesis, and cyanoamino acid metabolism were enriched commonly in the R_vs._Y, R_vs._P, and P_vs._Y comparison groups. The DEGs of the P_vs._Y comparison group were specifically enriched in tryptophan metabolism, pentose phosphate pathway, and isoflavonoid biosynthesis. Finally, the R_vs._P DEGs were specifically enriched in ubiquinone and other terpenoid—quinone biosynthesis, riboflavin metabolism, phenylalanine, tyrosine and tryptophan biosynthesis, fructose and mannose metabolism, flavone and flavonol biosynthesis, fatty acid elongation, carbon fixation in photosynthetic organisms, and anthocyanin biosynthesis (Figure 3D; Table S3).

*3.4. Differentially Accumulated Metabolites among Yellow, Red, and Purple Wolfberry Fruits*

In this study, the orthogonal partial least squares discriminant analysis (OPLS-DA) models were established and the 200 permutation cross validation tests were performed

by SMICA. As shown in Figure 4A–C, the OPLS-DA models' R2Y and Q2 values between the yellow, red, and purple groups indicated that the model had good stability, no over-fitting, and could be used to calculate the variable importance for the projection (VIP) of metabolites (Table S4; Figure 4A–C). A cluster heatmap of the differential metabolites among yellow, red, and purple wolfberry fruits was plotted and it was found that alkaloids, nucleotides, and amino acids and their derivatives, flavonoids, phenols, vitamins, organic acids, anthocyanins, and other types of compounds were differentially accumulated in yellow, red, and purple wolfberry fruits (Figure 4D). Interestingly, we found that purple wolfberries tended to accumulate more malvidin and petunidin 3,5-diglucoside, narirutin, naringin, prunin and phloridzin compared to red and yellow wolfberries; leucodelphinidin, eriodictyol, 6,8-di-C-glucoside apigenine, tricin o-saccharic acid, sakuranetin, apigenin 7-O-glucoside and C-hexosyl-tricetin O-pentoside tended to accumulate in yellow wolfberries; flavanones like naringenin, hesperidin, naringenin-7-O-glucoside tended to accumulate in red wolfberries (Table 3). Thus, these results show that flavonoids and anthocyanins are directly related to the color differences between wolfberry fruits.

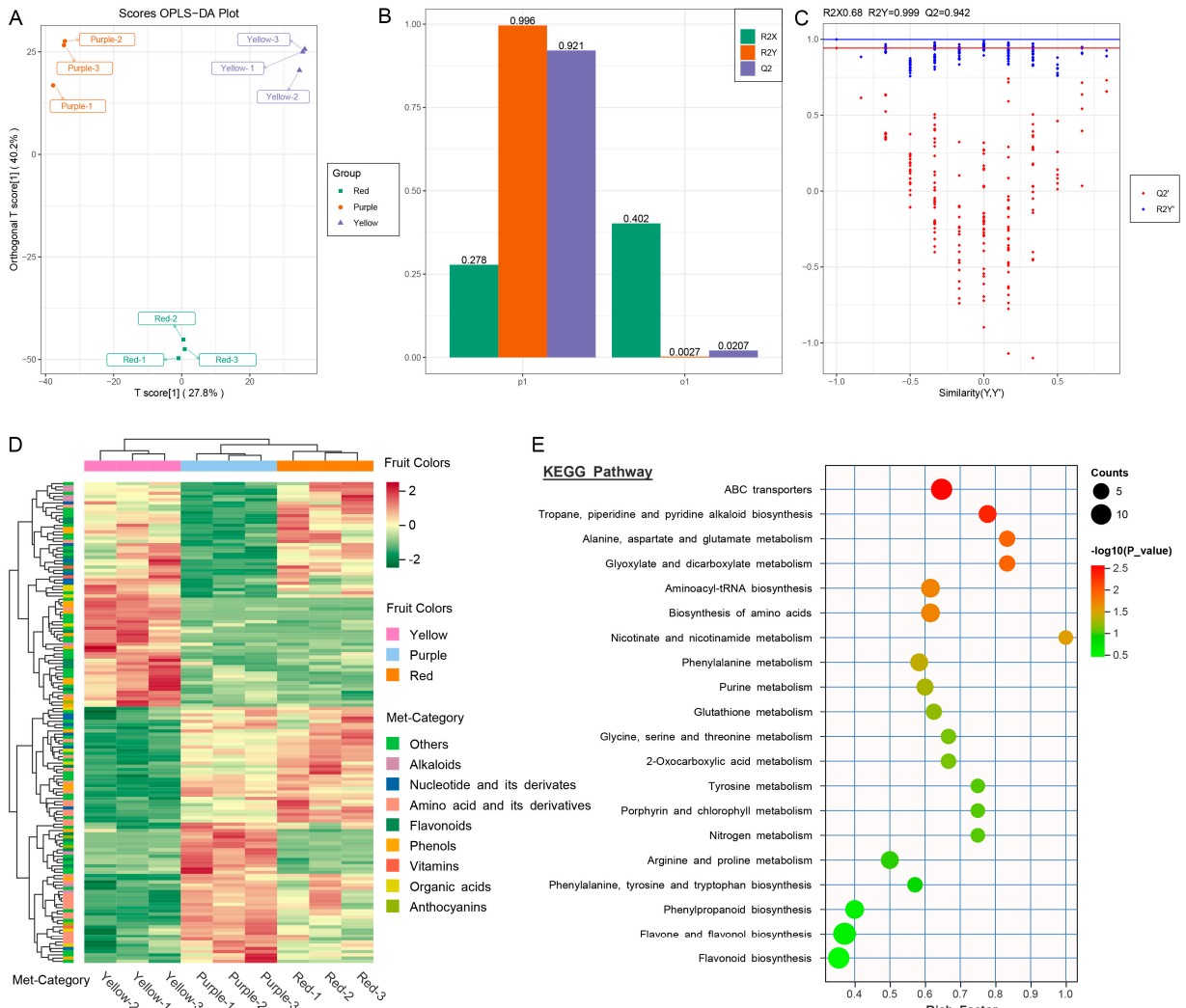

**Figure 4.** Statistical and functional enrichment analysis of differentially accumulated metabolites. (**A**) The OPLS-DA score plot. (**B**) The parameters of the OPLS-DA model. (**C**) Permutation test plot of OPLS-DA. (**D**) The hierarchical cluster heatmap of DAMS among yellow, red and purple fruits. (**E**) The babble lot of KEGG enrichment analysis of DAMS.

**Table 3.** Differentially accumulated flavonoids and anthocyanin metabolites in wolfberry fruits.

| Compounds | Class | Relative Content | | | VIP | *p* Value |
|---|---|---|---|---|---|---|
| | | Red | Purple | Yellow | | |
| Malvidin-Rut | Anthocyanins | $1.54 \times 10^4$ | $3.48 \times 10^5$ | $1.18 \times 10^4$ | 1.68 | $7.17 \times 10^{-6}$ |
| Leucodelphinidin-Hex | Anthocyanins | $4.54 \times 10^6$ | $4.33 \times 10^6$ | $6.14 \times 10^6$ | 1.55 | $1.72 \times 10^{-2}$ |
| Cyanidin-3-galactoside chloride | Anthocyanins | $1.75 \times 10^6$ | $3.76 \times 10^6$ | $6.57 \times 10^5$ | 1.85 | $9.11 \times 10^{-5}$ |
| petunidin 3,5-diglucoside | Anthocyanins | $1.34 \times 10^5$ | $8.32 \times 10^5$ | $7.83 \times 10^4$ | 1.77 | $7.82 \times 10^{-4}$ |
| Narirutin | Flavanone | $2.22 \times 10^4$ | $1.21 \times 10^5$ | $3.30 \times 10^4$ | 1.34 | $1.91 \times 10^{-4}$ |
| Naringin | Flavanone | $1.18 \times 10^4$ | $7.95 \times 10^5$ | $2.17 \times 10^4$ | 1.50 | $5.86 \times 10^{-5}$ |
| Naringenin 7-O-glucoside (Prunin) | Flavanone | $3.71 \times 10^5$ | $5.51 \times 10^7$ | $4.13 \times 10^5$ | 1.63 | $3.41 \times 10^{-6}$ |
| Naringenin | Flavanone | $1.82 \times 10^6$ | $5.92 \times 10^5$ | $1.54 \times 10^6$ | 1.47 | $3.78 \times 10^{-4}$ |
| Eriodictyol | Flavanone | $3.32 \times 10^3$ | $1.96 \times 10^4$ | $1.16 \times 10^5$ | 1.11 | $1.48 \times 10^{-3}$ |
| Hesperetin 7-rutinoside (Hesperidin) | Flavanone | $4.03 \times 10^8$ | $2.93 \times 10^8$ | $3.85 \times 10^8$ | 1.26 | $3.06 \times 10^{-2}$ |
| Naringenin-7-O-glucoside | Flavanone | $6.32 \times 10^5$ | $2.16 \times 10^5$ | $4.78 \times 10^4$ | 1.08 | $4.99 \times 10^{-4}$ |
| 6,8-di-C-glucoside Apigenine | Flavone | $6.77 \times 10^3$ | $7.26 \times 10^3$ | $2.26 \times 10^4$ | 1.38 | $4.03 \times 10^{-3}$ |
| Chrysoeriol 7-O-hexoside | Flavone | $6.88 \times 10^5$ | $4.23 \times 10^5$ | $7.12 \times 10^5$ | 1.38 | $4.60 \times 10^{-2}$ |
| Apigenin 7-O-glucoside (Cosmosiin) | Flavone | $7.61 \times 10^5$ | $1.68 \times 10^5$ | $7.43 \times 10^5$ | 1.60 | $5.63 \times 10^{-4}$ |
| Tricin O-saccharic acid | Flavone | $4.90 \times 10^5$ | $2.77 \times 10^5$ | $1.01 \times 10^6$ | 1.67 | $2.43 \times 10^{-2}$ |
| sakuranetin | Flavone | $1.17 \times 10^3$ | $1.13 \times 10^3$ | $3.49 \times 10^3$ | 1.34 | $2.87 \times 10^{-2}$ |
| Apigenin 7-O-glucoside | Flavone | $1.33 \times 10^6$ | $3.54 \times 10^5$ | $1.60 \times 10^6$ | 1.66 | $1.39 \times 10^{-2}$ |
| C-hexosyl-tricetin O-pentoside | Flavonoid | $6.19 \times 10^4$ | $6.49 \times 10^4$ | $2.31 \times 10^5$ | 1.47 | $9.02 \times 10^{-3}$ |
| Phloridzin | Flavonoid | $1.56 \times 10^5$ | $3.35 \times 10^5$ | $1.32 \times 10^5$ | 1.67 | $3.35 \times 10^{-4}$ |
| Pinobanksin | Flavonol | $2.81 \times 10^6$ | $1.02 \times 10^6$ | $2.44 \times 10^6$ | 1.48 | $6.13 \times 10^{-4}$ |

In order to understand the function of these different metabolites, KEGG pathway enrichment analysis was conducted. It was found that the DAMs among the three groups were significantly enriched in pathways such as ABC transporters, tropane, piperidine and pyridine alkaloid biosynthesis, alanine, aspartate and glutamate metabolism, glyoxylate and dicarboxylate metabolism, aminoacyl-tRNA biosynthesis, the biosynthesis of amino acids, nicotinate and nicotinamide metabolism, phenylalanine metabolism, purine metabolism, glutathione metabolism, and glycine, serine, and threonine metabolism. Additionally, it was found that many pathways were associated with flavonoids and anthocyanins. These include phenylalanine metabolism, phenylalanine, tyrosine and tryptophan biosynthesis, phenylpropanoid biosynthesis, and flavonoid, flavone, and flavonol biosynthesis (Figure 4E; Table S5).

### 3.5. Integrative Analysis of Transcriptome and Metabolome

To globally understand the regulation network of flavonoid and anthocyanins synthesis in wolfberry fruit, basing on the functional enrichment analysis results, the DEGs and DAMS which enriched in the flavonoid, flavone, and flavonol biosynthesis pathway were selected to conduct the transcriptomics and metabolomics integrative analysis (Tables S6 and S7). Aiming to discover signatory genes and metabolites involved in flavonoid and anthocyanins synthesis, data integration with O2PLS was performed on both datasets to reduce the noise and number of dominant correlations allowing for more confident biological interpretation and predictions. The O2PLS results of transcriptomics and metabolomics dataset revealed that flavonoids like hesperidin, trifolin, rutin, nicotiflorin, prunin, neohesperidin, and isotrifoliin A were strongly correlated with genes such as CCG033659(F3H), CCG036272(AOMT), CCG041018(ANS), CCG028824(DFR), and CCG025288(CHS) (Figure 5A,B). The Pearson correlation analysis and correlation network suggested that prunin showed significant positive correlation with most DEGs in flavonoid biosynthesis and flavone and flavonol biosynthesis, and isotrifoliin showed significant negative correlation with these genes (Figure 5C,D).

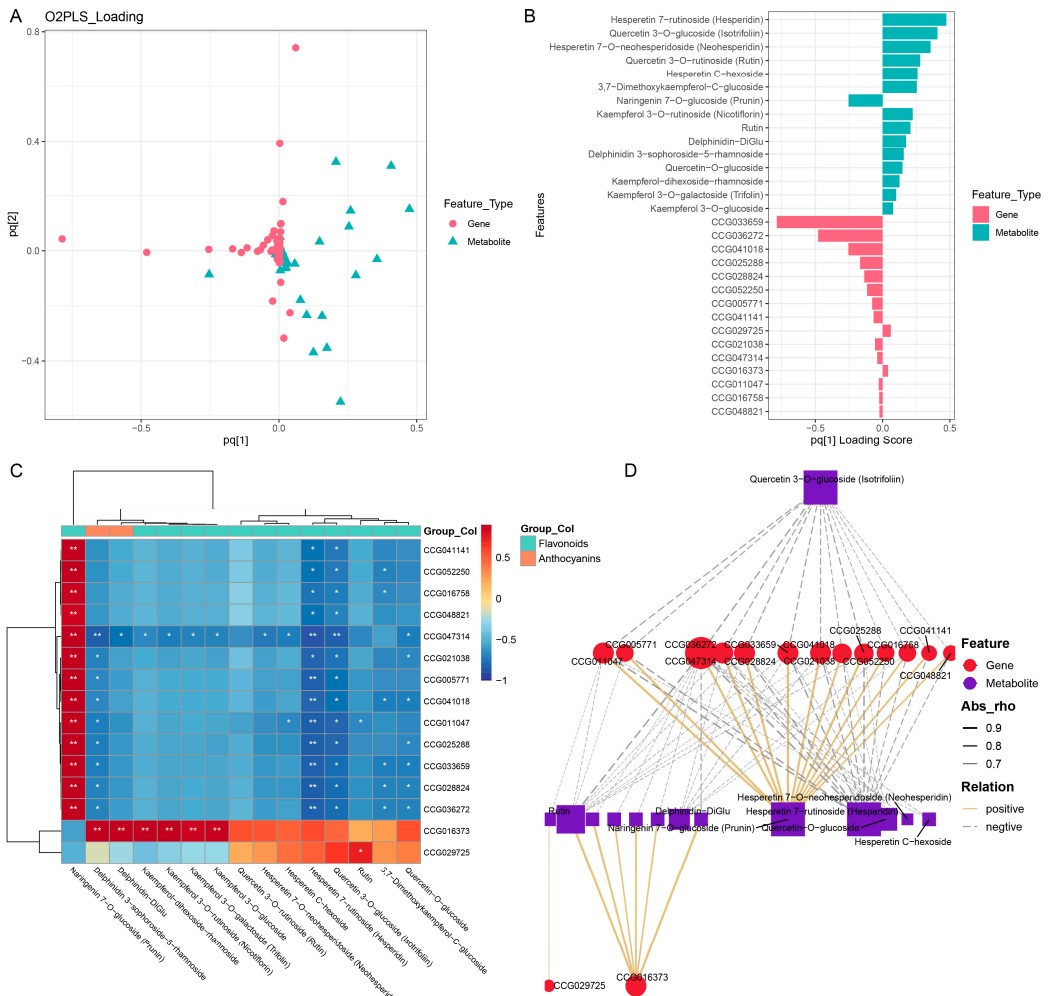

**Figure 5.** Integrated analysis of the DEGs and DAMS. (**A**) The O2PLS loading plot from the gene (p) and metabolite (q) loadings blocks of the DEGs and DAMs. Genes (circles) and metabolites (triangles) represent individual gene and metabolite loading values (**B**) The loading score barplot of the top 15 related metabolites and genes. (**C**) Hierarchical cluster heatmap of the Pearson correlation analysis of top 15 related metabolites and genes. "*" and "**" refer to Pearson correlation coefficients significance levels <0.05 and <0.01, respectively. (**D**) Correlation network diagram of the top 15 metabolites and genes.

## 4. Discussion

### 4.1. The Composition and Content Difference of Flavonoids Result in the Fruit Color Difference of Wolfberry

Metabolomics is an effective tool for measuring, identifying, and quantifying metabolites that have been used in various plant species [16,23]. In this study, the widely targeted metabolomic approach was adopted to investigate the metabolic changes in yellow, red, and purple wolfberry fruits. It is known that flavonoids, including flavones, isoflavones, flavanols, flavanones, and anthocyanins, play a vital role in plant tissue color formation [24–26]. The color of a fruit is essentially determined by its intrinsic substance and its content [27,28]. In our study, the metabolome results of three typical color differences of *Lycium barbarum* showed that red, yellow and purple wolfberry had significant differences in the accumulation of metabolites. Many DAMs between the three different colors are flavonoid compounds, such as flavonols and anthocyanins. And the accumulation pattern of flavonoids of wolfberries in different colors was different: anthocyanins tended to accumulate in purple wolfberry, for example, malvidin and petunidin 3,5-diglucoside; flavones tended to accumulate in yellow wolfberry, such as 6,8-di-C-glucoside apigenine,

tricin O-saccharic acid and sakuranetin; flavanones like naringenin, hesperidin, naringenin-7-O-glucoside tended to accumulate in red wolfberries. When anthocyanins, flavonoids, and flavonols are mixed according to different proportions and contents, they will show different color characteristics [29], which means that the differences in flavonoid composition and content are the material basis for the color difference of wolfberry fruits. Anthocyanins are important pigmentation pigments in plant, with a color gamut ranging from red to purple, for example, malvidin and petunidin which accumulate in purple wolfberry appear bluish-red in color [25,26,29,30]. While yellowish flavonoids, which play a key role in plant coloring [31], such as apigenine [32], sakuranetin, naringenin [33], and hesperidin accumulate in red and yellow wolfberries. In summary, we hypothesized that purple wolfberries could synthesize or accumulate more bluish anthocyanins and appear dark purple, while yellow and red wolfberries may contain more yellowish flavonoids and appear yellow and red. Therefore, *Lycium barbarum* cultivars with different colors can be obtained by controlling the biosynthesis and accumulation of anthocyanins or other flavonoids in breeding.

### 4.2. The Expression Differences of Flavonoid Biosynthesis Genes Reshape the Carbon Flow in Flavonoid Metabolic Pathways

In this study, KEGG pathway enrichment analysis showed that phenylalanine metabolism, phenylalanine, tyrosine and tryptophan biosynthesis, phenylpropanoid biosynthesis, and flavonoid, flavone, and flavonol biosynthesis pathways were enriched in the DAMs among yellow, red, and purple wolfberry fruits (Figure 4E). Anthocyanidin biosynthesis belongs to a branch pathway of flavonoids that starts from phenylalanine (Figure 6). A variety of enzymes participate in the biosynthesis of anthocyanins in higher plants [34]. In our study, the transcriptomics results showed that phenylpropanoid biosynthesis, phenylalanine metabolism, flavonoid biosynthesis, isoflavonoid biosynthesis, phenylalanine, tyrosine and tryptophan biosynthesis, flavone and flavonol biosynthesis, and anthocyanin biosynthesis were enriched in the DEGs between yellow, red, and purple wolfberry fruits (Figure 3C,D).

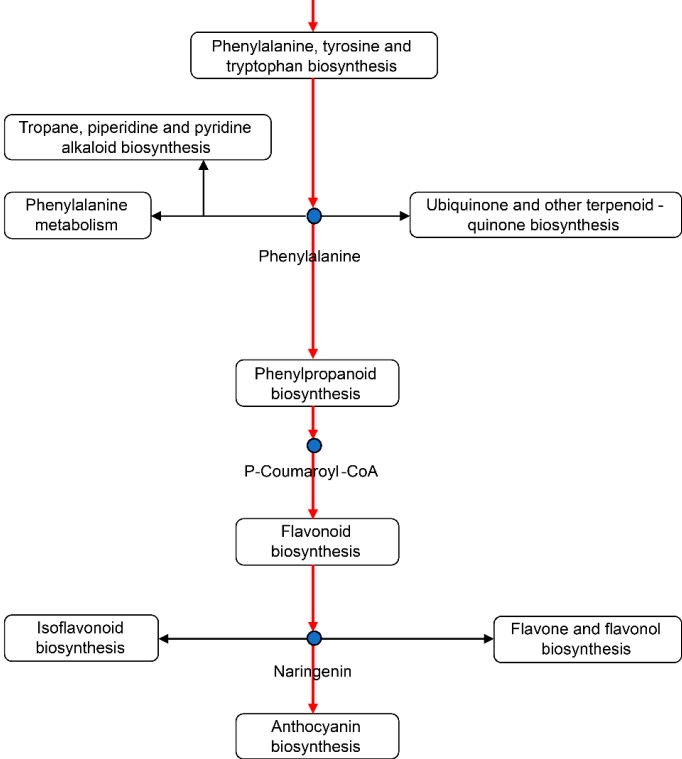

**Figure 6.** Schematic diagram of the anthocyanin biosynthesis pathway [9,10].

Chalcone synthase (CHS) is a key enzyme in the flavonoid biosynthesis pathway [35,36]. Previous studies revealed that CHS is a key regulatory protein for anthocyanin biosynthesis in red nectarines and peaches, and the downregulation of the CHS gene could switch the anthocyanin pathway to chlorogenic acid synthesis [37]. In this study, a total of 9 CHS genes were identified by RNA-sequencing, among which CCG011047 and CCG025288 were differentially expressed between red, yellow, and purple wolfberries, and purple wolfberry showed the highest expression level of these CHS genes. Flavanone 3-hydroxylase (F3H), dihydroflavonol-4-reductase (DFR), and anthocyanin synthase (ANS) are key enzymes in the synthesis of flavonols and anthocyanins [10]. Our study revealed that an ANS-coding gene (CCG041018), two F3H-coding genes (CCG019697 and CCG033659), and one DFR-coding genes (CCG028824) were differentially expressed between red, yellow, and purple wolfberries. Interestingly, most of these genes also expressed high levels in the purple wolfberries. Metabolomics showed that purple wolfberry synthesized and accumulated more bluish anthocyanins, and transcriptomics showed that the key enzyme genes of anthocyanins biosynthesis remained at higher expression levels in purple wolfberry than red and yellow ones, suggesting that controlling or modifying the key enzyme genes of anthocyanin synthesis could control the fruit color of wolfberry.

Besides these enzymes above, a lot of studies have shown that MYB and bHLH transcription factors also play important roles in plant coloring [23,24,38–41]. For instance, Wang et al. found a promising candidate flavonoid for regulating transcription factor LrMYB1, which can induce ectopic flavonoid accumulation in in *L. ruthenicum* [42]. Lin Tang et al. found that the abnormal expression of LrAN1b, a bHLH transcription factor, results in a white fruit phenotype in *Lycium ruthenicum* [18]. In our transcriptome study, it was shown that several MYB and bHLH genes were differentially expressed in different colored wolfberry fruits and particularly highly expressed in the fruits of purple wolfberry. The transcriptomics results are consistent with the metabolomics results. Therefore, it can be speculated that MYB or bHLH transcription factors could regulate flavonoids biosynthesis related genes to control the composition of different flavonoids or anthocyanins in wolfberry fruits and result in the different fruit colors.

## 5. Conclusions

In this study, metabolomics and transcriptomics analysis were integrated to elucidate the regulatory mechanisms of flavonoids biosynthesis in red, yellow, and purple wolfberry. Metabolomics analysis revealed that flavonoids, phenols, vitamins, anthocyanins, and other compounds differentially accumulated among yellow and purple wolfberry fruits. And the flavonoid composition and content of wolfberries of different colors were different: the bluish anthocyanins, malvidin and petunidin tended to accumulate in purple wolfberry, while red and yellow wolfberries tended to accumulate more yellowish flavonoids. The different flavonoid accumulation patterns may result in the different fruit colors of wolfberry. Transcriptome analysis showed that numerous flavonoid-synthesis-related genes such as CHS, F3H, ANS, and DFR, and several MYB and bHLH genes were differentially expressed among wolfberry fruits of different colors: most of these genes were more highly expressed in purple wolfberries than in red and yellow wolfberries. The different expression profiles of flavonoid biosynthesis genes and MYB and bHLH genes could reshape flavonoid metabolic pathways and regulate the fruit color of wolfberry. In conclusion, the different flavonoid accumulation patterns may result in the different fruit colors of wolfberry, and the MYB or bHLH transcription factors could regulate the expression of flavonoid-biosynthesis-related genes to change the composition of flavonoids or anthocyanins in wolfberry fruits and result in various fruit colors. These findings provide new insights into the underlying molecular mechanisms of the fruit color differences in wolfberry and provide new ideas for molecular breeding of wolfberry.

**Supplementary Materials:** The following supporting information can be downloaded at: https://www.mdpi.com/article/10.3390/agronomy13071926/s1, Table S1: All differentially expressed genes between red, purple, and yellow wolfberry fruits; Table S2: GO functional enrichment results of differentially expressed genes between red, purple, and yellow wolfberry fruits; Table S3: KEGG functional enrichment results of differentially expressed genes between red, purple, and yellow wolfberry fruits. "*", "**", "***" and "****" refer to significance levels <0.05, <0.01, <0.005 and <0.001, respectively; Table S4: All different metabolites between red, yellow, and purple wolfberry fruits; Table S5: KEGG functional enrichment results of different metabolites between red, purple, and yellow wolfberry fruits; Table S6: DEGs related to flavonoids biosynthesis in wolfberry fruits; Table S7: Different metabolites involved in flavonoids biosynthesis between wolfberry fruits.

**Author Contributions:** Conceptualization, J.Z. and K.Q.; methodology, L.D. and B.Z.; software, G.D. and X.H.; validation, G.D., X.H., X.Z. and X.L.; formal analysis, L.D., B.Z., T.H. and X.L.; investigation, G.D., X.H., X.Z., T.H. and X.L.; resources, G.D., J.Z. and K.Q.; data curation, L.D., B.Z., X.Z. and T.H.; writing—original draft preparation, L.D. and B.Z.; writing—review and editing, J.Z. and K.Q.; visualization, L.D. and B.Z.; supervision, J.Z. and K.Q.; project administration, G.D. and K.Q.; funding acquisition, J.Z. and K.Q. All authors have read and agreed to the published version of the manuscript.

**Funding:** This research was supported by the Key Research & Development Program of Ningxia Hui Autonomous Region (2021BEF02002), the Innovative Research Group Project of Ningxia Hui Autonomous Region (2021AAC01001), the National Natural Science Foundation of China (32060359), and the Ningxia Hui Autonomous Region key research and development projects (2022BBF02003).

**Data Availability Statement:** The raw data of the transcriptomes have been deposited to the GenBank database with BioProject ID PRJNA953064. All other data generated or analyzed in this paper are included in the manuscript and Supplementary Files.

**Conflicts of Interest:** The authors declare no conflict of interest.

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
