# Peer review of "Integrated Analysis of Transcriptome and Metabolome Reveals New Insights into the Molecular Mechanism Underlying the Color Differences in Wolfberry (Lycium barbarum)"

_agronomy, doi:10.3390/agronomy13071926_

Round 1

Reviewer 1 Report

The authors propose a manuscript titled “Integrated analysis of transcriptome and metabolome reveals new insights into the molecular mechanism underlying the color differences in wolfberry (Lycium barbarum)”.

I suggest the following changes:

The species should be written in italics.

Discussion should be expanded, as well as the conclusions.

Minor editing of English language required

Author Response

The species should be written in italics.

Reply: Species in the paper have been written in italics

Discussion should be expanded, as well as the conclusions.

Reply: Discussion and conclusions have been expanded.

Comments on the Quality of English Language

Minor editing of English language required

Reply: We have edited English language of this paper and we will further improve the language with the help of professional English language editing agencies.

Reviewer 2 Report

The manuscript entitled “Integrated analysis of transcriptome and metabolome reveals 2 new insights into the molecular mechanism underlying the 3 color differences in wolfberry (Lycium barbarum)” by Duan et al has analyzed transcriptome and metabolome of three red, yellow and purple wolfberry lines to understand the mechnisim of color differences. The manuscript is interesting for researchers in plant specialized metabolism and breeding in wolfberry. However, I have few comments and concerns:

Line 68—69, how many biological replications for red, yellow and purple lines?

Line 115, add (DMS) after significantly differential metabolites.

Line 130-131, provide citation for Lycium barbarum reference genome sequence.

Line 134, what is the software used for selected DEGs? How does the Student T test adjust p value e.g. the false discovery rate (FDR)?

 Figure 2 C, D, need to provide P value for the Pearson correlation coefficient.

 Table 2 is not clear. How do authors selected DEGs related to flavonoids biosynthesis pathway? Table2 needs to provide folder changes, p value and gene annotations of listed DEGs.

Line 227, authors need to provide specific chemical compounds of flavonoids and anthocyanins. A table has all listed chemical compounds or DAMs will be very helpful.

Line 279, I do not find results of accumulation of metabolites. Authors need to provide a table to list all identified metabolites (DAMs).

 Line 317-319, I do not understand the sentences. It seems a Reviewer’s comments.

 Line 351, I do not find any supplementary files.

Author Response

Line 68—69, how many biological replications for red, yellow and purple lines?

Reply: All experimental samples were repeated three times.

Line 115, add (DMS) after significantly differential metabolites.

Reply: We have made corresponding revision in the paper.

Line 130-131, provide citation for Lycium barbarum reference genome sequence.

Reply: We have added the citation for Lycium barbarum reference genome sequence.

Line 134, what is the software used for selected DEGs? How does the Student T test adjust p value e.g. the false discovery rate (FDR)?

Reply: We used the Q value < 0.05 for select DEGs. And we have made corresponding revision in the paper.

Figure 2 C, D, need to provide P value for the Pearson correlation coefficient.

Reply: Figure 2 C, D have been replotted and the significance for the Pearson correlation coefficient was displayed as “*” and “**” in the figure.

Table 2 is not clear. How do authors selected DEGs related to flavonoids biosynthesis pathway? Table2 needs to provide folder changes, p value and gene annotations of listed DEGs.

Reply: We have added descriptions of the selecting process of DEGs and DAMs related to flavonoids biosynthesis pathway in corresponding positions in the paper, and the table has been improved and supplemented with an additional material, table S1, which contains complete information such as folder changes, p value and gene annotations.

Line 227, authors need to provide specific chemical compounds of flavonoids and anthocyanins. A table has all listed chemical compounds or DAMs will be very helpful.

Reply: We added Table 3 to show flavonoids in the article, and table S4 containing all DAMs information was provided in the supplementary materials.

Line 279, I do not find results of accumulation of metabolites. Authors need to provide a table to list all identified metabolites (DAMs).

Reply: We added Table 3 to show flavonoids in the article, and table S4 containing all DAMs information was provided in the supplementary materials.

Line 317-319, I do not understand the sentences. It seems a Reviewer’s comments.

Reply: We have made corresponding revision in the paper.

Line 351, I do not find any supplementary files.

Reply: We have added words in supplementary files section, and provided 7 supplementary files, which contains most of the data and results of data analysis.

Reviewer 3 Report

Abstract

Add some of results transcriptome and metabolome which are best.

Introduction

Write a little about models, importance of study interims of future benefits.

Materials and methods

Sources of plant material, you mention lines, give details of lines in terms of crosses.

you research is simple for well-known plant materials; cross line research will be different.

How about number of samples. too small for lines.

There is no cross match of these three colors in terms of analysis.

Results

How about fig. 1 confirm this with color scheme.

Give some supplemental materials to understand genetics, give original graphics of HPLC etc. gel pics

Reference:

Correct in one format.  

 Minor editing of English language required

Author Response

Abstract

Add some of results transcriptome and metabolome which are best.

Reply: We revised the Abstract to add key results for the metabolome and transcriptome.

Introduction

Write a little about models, importance of study interims of future benefits.

Reply: In introduction, we have added some words of research purpose and significance, and forecast the future benefits.

Materials and methods

Sources of plant material, you mention lines, give details of lines in terms of crosses.

you research is simple for well-known plant materials; cross line research will be different.

How about number of samples. too small for lines.

There is no cross match of these three colors in terms of analysis.

Reply: In this study, we only selected these three representative wolfberry cultivars to preliminarily explore the differences in gene expression and metabolite accumulation among them. All experimental samples were repeated three times.

Results

How about fig. 1 confirm this with color scheme.

Give some supplemental materials to understand genetics, give original graphics of HPLC etc. gel pics

Reply: The original intention of this study was to preliminarily explore the effects of differences in gene expression and metabolite accumulation on the fruit color of Wolfberry, without considering the relationship between the genetics of the materials used. This is the deficiency.

Reference:

Correct in one format.  

Reply: We re-did the references citations and lists with endnote software in MDPI reference style.

Comments on the Quality of English Language

Minor editing of English language required

Reply: We have edited English language of this paper and we will further improve the language with the help of professional English language editing agencies.

Round 2

Reviewer 1 Report

The authors have provided a revised version addressing most of my comments, however, I consider it necessary to add more references in the discussion section.

Minor editing of English language required

Author Response

The authors have provided a revised version addressing most of my comments, however, I consider it necessary to add more references in the discussion section.

Reply:We have added some new references to some ideas or conclusions in the discussion section.